

# Evolution of genes involved in feeding preference and metabolic processes in Calliphoridae (Diptera: Calyptratae)

Gisele Antoniazzi Cardoso[1], Marco Antonio Tonus Marinho[2], Raquel Dietsche Monfardini[1], Ana Maria Lima de Azeredo Espin[3] and Tatiana Teixeira Torres[1]

[1] Department of Genetics and Evolutionary Biology, University of Sao Paulo, Sao Paulo, Brazil
[2] Department of Biology, Faculty of Philosophy, Science, and Letters, University of Sao Paulo, Ribeirao Preto, Sao Paulo, Brazil
[3] Department of Genetics, Evolution, and Bioagents, State University of Campinas, Campinas, Sao Paulo, Brazil

## ABSTRACT

**Background**. The genotype-phenotype interactions among traits governing feeding preference are of fundamental importance to behavioral genetics and evolutionary biology. The genetic basis of behavioral traits has been explored in different taxa using different approaches. However, the complex nature of the genetic mechanisms undergirding behavior is poorly understood. Here, we present an evolutionary study of candidate genes related to parasitism in Calliphoridae (Diptera: Calyptratae). Closely related species in this family exhibit distinct larval feeding habits, most notably necro-saprophagy and obligate parasitism.

**Methods**. To understand the genetic and molecular bases underlying these habits, expression levels of eight candidate genes for feeding behavior—*Cyp6g2, foraging, glutamate dehydrogenase, Jonah65aiv, Malvolio, PGRP-SC2, RPS6-p70-protein kinase,* and *smooth*—were measured in four species using qPCR. Moreover we used expression values and sequence information to reconstruct the relationship among species and the $d_N/d_S$ rate to infer possible sites under selection.

**Results**. For most candidate genes, no statistically significant differences were observed, indicating a high degree of conservation in expression. However, *Malvolio* was differentially expressed between habits. Evolutionary analyses based on transcript levels and nucleotide sequences of *Malvolio* coding region suggest that transcript levels were correlated to feeding habit preferences among species, although deviations under a strictly neutral model were also observed in statistical tests.

**Discussion**. *Malvolio* was the only gene demonstrating a possible connection to feeding habit. Differences in gene expression may be involved in (or be a result of) the genetic regulation of Calliphoridae feeding habit. Our results are the first steps towards understanding the genetic basis and evolution of feeding behavior in Calliphoridae using a functional approach.

Corresponding author
Tatiana Teixeira Torres,
tttorres@ib.usp.br

## INTRODUCTION

One of the most interesting and complex phenotypic traits that challenges molecular biologists is behavior. Behavioral traits are very plastic and can be influenced by multiple genes, organism needs, and environmental factors. Despite the complexity of behavioral genetics, previous studies were able to successfully identify genes underlying specific behaviors, especially in model species as *Drosophila melanogaster* (*Sokolowski, 2001*). A large number of genes have been described to play a role in behaviors including aggressiveness (*fruitless*), courtship dynamics (*courtless*), and circadian rhythms (*period*) (*Greenspan & Ferveur, 2000*; *Ryner et al., 1996*; *Waddell & Quinn, 2001*). Understanding the genetic complexity and evolution of these traits is of fundamental importance to elucidate the behavioral influences on ecological diversification, adaptation, and speciation. Variations in gene expression patterns are responsible for a large proportion of phenotypic differences among species (*Enard et al., 2002*; *King & Wilson, 1975*). These differences may be due to mutations in coding or regulatory regions or changes in the chromatin structure that affect accessibility of transcriptional machinery to the genetic template (*Enard et al., 2002*; *Yang et al., 2006*). Traditionally, researchers have focused on coding regions, but there is now a growing number of studies investigating evolution through differentiation of gene expression patterns. Many of these studies have attempted to address the question of how phenotypic differences among closely related species are connected with differences in gene expression. In Darwin finches, for example, differences in expression patterns of the *bone morphogenetic protein 4* (*Bmp4*) and *calmodulin* (*CaM*) genes have been shown to be involved in beak shape variation among closely related species, a trait that directly influences diet and exploitation of food resources (*Abzhanov et al., 2006*; *Abzhanov et al., 2004*). In addition, regulatory differences in the *Pituitary homeobox 1* gene (*Pitx1*) between marine and fresh water populations of *Gasterosteus aculeatus* have been shown to be associated with the progressive reduction of pelvic spines, which have an adaptive function in defense against predation (*Shapiro et al., 2004*). These differences in regulatory regions are to a large extent responsible for the morphological variation on which natural selection can act.

In this context, the Calliphoridae family (Diptera: Calyptratae) constitutes an ideal system to investigate differential patterns of gene expression related to the evolution of feeding behavior. Most species in this family exhibit two main feeding habits in their immature stage: necro-saprophagous and parasitic. Necro-saprophagous larvae feed on decaying organic matter such as feces, animal carcasses, and other decomposing organic substrates (*Zumpt, 1965*; *De Jong, 1976*). These larvae are not able to develop in living tissues, but may be found feeding on necrotic tissues of wounded animals (*Stevens & Wall, 1997*). The parasitic habit refers to larvae that can, in an obligate or facultative manner, feed on living tissues of warm-blooded vertebrates, causing infestations called myiasis (*Zumpt, 1965*; *Hall & Wall, 1995*). Unlike other parasites, larvae of Calliphoridae species are only capable of initiating an infestation in open wounds or body orifices (mucosal tissues). Once the host is infested, the larvae can cause extensive tissue damage, usually a cavernous wound (*Hall & Wall, 1995*).

The necro-saprophagous habit of blowflies seems to comprise the ground-plan for the group and, according to the current phylogenetic hypothesis for this family, obligate parasitism appears to have evolved independently at least three times (*Stevens & Wallman, 2006*). In the most parsimonious scenario, the primitive necro-saprophagous habit evolved to an obligate parasitic habit through the appearance of opportunistic infestations of living animals, which were weak or with partly necrotic wounds (*Erzinçlioglu, 1989*; *Zumpt, 1965*). Two Calliphoridae genera in the subfamily Chrysomyinae, *Chrysomya* and *Cochliomyia*, contain both obligate parasites and necro-saprophagous species, allowing the comparison of different feeding habits among closely related species.

Here, we use a candidate gene approach as a first step to investigate the evolution of feeding habits in Calliphoridae. The qPCR analyses of transcripts from eight candidate genes in four closely related species, *Chrysomya albiceps*, *Chrysomya megacephala*, *Cochliomyia macellaria* (all with necro-saprophagous larvae), and *Cochliomyia hominivorax* (whose larvae are obligate parasites of mammals) were used to determine genetic associations with different feeding habits and the types of selection potentially influencing gene expression. Furthermore, the results from the gene expression analysis correlated with the results from molecular evolution and sequence analyses of the candidate genes.

## MATERIALS AND METHODS

Fly collection and rearing: adult flies of the necro-saprophagous Calliphoridae species were collected using a hand net and decaying meat or fish as baits. Adults of *Ch. albiceps, Ch. megacephala*, and *Co. macellaria* were collected in three cities in the São Paulo state in Brazil: Campinas, Rio Claro, and Sorocaba, respectively. *Co. hominivorax* larvae were collected from infested wounds in cattle breeding farms in Caiapônia, Goiás, Brazil. Larvae of the four species were reared at $30° \pm 5°$. *Co. hominivorax* larvae were maintained in a medium consisting of fresh ground meat supplemented with bovine blood and water (2:1). The necro-saprophagous larvae were fed on euthanized rats donated by the Multidisciplinary Center for Biological Research in the Laboratory of Animal Science (CEMIB, Unicamp). Mature larvae of the four species were allowed to pupate in sawdust. Adults were maintained in cages ($34 \times 50 \times 26$) at 25 °C and fed with a diet composed of dried milk, sugar and brewer's yeast (3:2:1).

Candidate gene selection and primer design: candidate genes were selected from previous studies in Diptera (Table S1). To expand our search, the Gene Ontology (GO) database was also used (available at http://www.geneontology.org). Sequences of the selected genes were obtained from Flybase (http://flybase.org/). The sequences of the twelve *Drosophila* species with whole genome sequenced (*Clark et al., 2007*) were aligned against all *Co. hominivorax* transcriptome contigs (*Carvalho, Azeredo-Espin & Torres, 2010*) to identify corresponding orthologs (50% of identity at least in 40% of the sequence). Identified contigs were aligned against the BLAST database repository (nt database) using blastn with default parameters (*Altschul et al., 1990*) to search for orthologs in closely related species. Primers for each gene were then designed in conserved regions among the selected orthologs with Primer3 (*Rozen & Skaletsky, 2000*) considering the respective criteria: melting temperatures 55–60 °C, primers lengths 15–22 bp and amplicon lengths 100–150 bp.

RNA isolation and complementary DNA (cDNA) synthesis: RNA extractions were performed on two separate generations of early 3rd instar (feeding) larvae, and newly emerged adult males and females of each species (*Ch. megacephala Ch. albiceps*, *Ch. megacephala*, *Co. hominivorax*, and *Co. macellaria*). Total RNA was extracted from 10 individuals from each generation using TRIzol® reagent (Invitrogen, USA) according to the manufacturer's protocol. For subsequent analyses, individuals of the same generation were combined into two RNA samples, each containing five individuals, for a total of four biological replicates.

All RNA extractions were treated with TURBODNase (Ambion® Life Technologies, ThermoFisher, MA USA) to degrade any remaining genomic DNA. DNase was inactivated by heating samples at 75° for 10 min followed by the addition of EDTA to a final concentration of 2.5 mM. After DNAse treatment, a PCR was performed using *Rp49* primers (as described in the Amplification section) to test the treated samples.

Total RNA was quantified with a Qubit® fluorometer (Invitrogen™, ThermoFisher, MA USA) using the Qubit RNA® assay kit. Synthesis of cDNA was performed using the First Strand cDNA Synthesis Kit (Thermo Fisher Scientific, Waltham, MA USA), following the manufacturer's protocol and using 0.5 μg of total RNA as template.

PCR amplification of selected genes: to test the candidate gene primers (Table S2), a PCR was performed using the cDNA of the four Calliphoridae species as template. PCR reactions were conducted in in a GeneAmp PCR system 9700 (Applied Biosystems, Life Technologies) for a final volume of 20 μL containing 2.0 mM MgCl$_2$, 0.6 μM of each primer, 200 μM dNTPs, and 1 U *Taq* DNA polymerase (Thermo Scientific, MA, USA). Reaction conditions consisted of an initial denaturing step of 3 min at 94 °C, followed by 35 cycles of 50 s at 94 °C, 30 s at 60 °C, and 30 s at 72 °C, and a final extension step at 72 °C for 5 min.

Quantitative PCR (qPCR): qPCR were performed in a final volume of 12.5 μL using the SYBR Green PCR Master Mix (Applied Biosystems, Foster City, CA, USA) according to manufacturer's instructions. In each reaction, 1 μL of cDNA was used and primers were added to a final concentration of 200 nM. The qPCRs were run in two technical replicates to assess intra-assay variation on an ABI 7500 Fast Real Time PCR System (Applied Biosystems, Foster City, CA, USA) using the following cycling conditions: 2 min at 50 °C, 10 min at 95 °C, and 40 cycles consisting of 15 min at 95 °C and 60 s at 60 °C. After the 40th cycle, the temperature was gradually increased from 60 °C to 95 °C (1 °C per minute for 35 min) to obtain a dissociation curve. This step was performed to confirm the formation of a single product during the reaction. A five point standard curve was used to access the efficiency of each primer. Efficiency values were calculated according to the $E = 10^{1/\text{slope}} - 1$ (*Higuchi et al., 1993*).

To determine the relative steady-state mRNA levels (which we will refer to as gene expression levels), the raw expression values, the cycle thresholds (*Ct*s), were normalized using genes with stable expression among different conditions/species. The efficiency of the amplification reactions were also measured and used for the normalization. This step was done to allow the comparison among different species (*Wong & Medrano, 2005*). The mean normalized expression (MNE) formula proposed by *Simon (2003)*

was used to calculate the relative gene expression values. MNE is the ratio of the mean of the reference and target genes cycle thresholds ($Ct$s), including efficiency values: $\text{MNE} = (E_{\text{reference}})^{\text{Ct reference}}/(E_{\text{target}})^{\text{Ct target}}$. All mRNA levels were normalized using the mean of the reference genes *Gapdh* and *Rp49*, both validated in a previous study (*Cardoso et al., 2014*).

Multivariate analyses: statistically significant differences ($\alpha = 0.01$) in steady-state mRNA levels between the studied species and different feeding habits were determined using two-way Analyses of Variance (ANOVA) (species × feeding habit). The MNE values were used as input data for all statistical analyses. All tests were performed in the statistical package R (*Team, 2008*). Genes differently expressed among species were candidates to be under neutral evolution or positive selection and genes with no significant difference were candidates to be under purifying selection.

Gene expression analyses: to test the hypothesis that species divergence in mRNA expression evolves neutrally, pair-wise differences in mRNA transcript levels were compared with a matrix of pair-wise genetic distances using linear regression analyses and the Pearson's correlation index.

Differences in mRNA levels were calculated using the square difference between each species pair. Pair-wise genetic distances among given species were based on a published concatenated dataset of COI, 16S, 28S, and ITS2 sequences (*Marinho et al., 2012*), and estimated using the Maximum Composite Likelihood method (with the TN93+G substitution model) in MEGA 5 (*Tamura et al., 2011*).

Gene expression trees (clustering analyses): pair-wise differences (absolute values) in expression levels among species, sexes, and developmental stages for each candidate gene were used to generate distance matrices for clustering analyses. These matrices were then analyzed using FastME software (*Desper & Gascuel, 2002*), which computes distance trees using the Minimum Evolution principle. Cluster diagrams (expression trees) were estimated separately for sex and developmental stage and with combined data. The resulting trees were then compared with the known phylogenetic relationships among these species, used as a null hypothesis for the observed pattern of gene expression data.

Sanger sequencing of candidate genes: five of the candidate genes (*for*, *Gdh*, *Jon65aiv*, *Mvl*. and *S6k*) showing the most promising results in the expression analyses were selected for further investigation using coding sequence data. Primers for PCR-amplification and sequencing of these gene regions were designed (Table S3), using the previously synthesized cDNA as a template.

PCR reactions of coding regions were performed in a 25 µL final volume, containing 0.45 µM of each primer, 200 µM dNTPs, and 1.25 U of DreamTaq DNA polymerase (Thermo Scientific, MA USA). Reaction conditions comprised an initial denaturing step of 3 min at 92°, followed by 30 cycles of 30 s at 95 , 30 s at specific primers' annealing temperatures (Table S3) and 30 s at 72°, with a final step of extension at 72° for 5 min.

PCR products were purified with Exonuclease I and FastAP Alkaline Phosphatase (ExoFap, Thermo Scientific) and sequenced in an ABI 3730 DNA Analyzer Sanger platform (Applied Biosystems, Foster City, CA, USA) using the BigDye® Terminator v3.1 Cycle Sequencing Kit (Life Technologies, Carlsbad, CA, USA).

Phylogenetic analyses (sequence dendrograms): sequences of the five genes obtained in the previous section were used to infer phylogenetic relationships among the species under study. Sequence alignments were generated using the module G-INS-I of MAFFT v 7.149 software (*Katoh et al., 2002*; *Katoh & Standley, 2013*) and resultant alignments were checked for out-of-frame indels and premature stop codons using the "Alignment Explorer" tool available in MEGA 6 software (*Tamura et al., 2013*). Maximum-likelihood (ML) and Bayesian Inference (BI) trees were inferred using GARLI v2.0 (*Zwickl, 2006*) and MrBayes v3.2.3 (*Ronquist et al., 2012*), respectively. In both analyses, the substitution model was estimated using jModelTest v2.1.5 (*Darriba et al., 2012*), and the best-fit model was selected for analyses. Resulting phylogenies were compared with the known phylogenetic relationships among the species, the latter of which were used as the null hypothesis for primary sequence evolution (i.e., neutral if coding sequences have evolved following lineage diversification; selection if there are forces constraining evolution based on larval alimentary habit).

Moreover, a maximum-likelihood (ML) tree was constructed for the *Mvl* gene (as described above) using the sequences of eight flies with different feeding habits: necro-saprophagous (*Ch. albiceps*, *Ch. megacephala*, *Co. macellaria*, and *Lucilia sericata*), obligate myiasis-causing parasites (*Co. hominivorax*, *Dermatobia hominis*, and *Oestrus ovis*), facultative myiasis-causing parasites (*Lucilia cuprina*), obligate hematophagous parasites (*Glossina morsitans*), and generalists (*Musca domestica*).

Coding sequence evolution: to perform a quantitative evaluation of the neutral hypothesis of sequence evolution, the Tajima's Relative Rates Test (*Tajima, 1993*) was conducted as implemented in MEGA v.6 (*Tamura et al., 2013*). This test provides a statistical framework to compare the evolutionary rates of two ingroup sequences in relation to a third (outgroup) sequence, and checks for deviations from the molecular clock hypothesis. For these analyses, the two *Cochliomyia* species (one necro-saprophagous and one obligate parasite) were defined as the ingroup taxa, and both *Chrysomya* species were tested as the outgroup.

In addition, a maximum-likelihood method was used to estimate the ratio of the rate of non-synonymous (non-silent) substitutions per site to the rate of synonymous (silent) substitutions per site ($d_N/d_S$, *Ka/Ks*, or $\omega$) for *Mvl* and to obtain the likelihoods ratio, using as input a neighbor joining (NJ) phylogeny. The codeml program from the PAML v.4 package (*Yang, 2007*) was used to test five models: (a) the null expectation from a neutral model with a fixed $\omega = 1$ (neutral model); (b) average $\omega$ for the whole tree (model 1); (c) specific $\omega$ for *Co. hominivorax* and an average $\omega$ for all other species (model 2); (d) an average $\omega$ for all parasites (*Co. hominivorax*, *D. hominis*, *G. morsitans*, *L. cuprina*, and *O. ovis*) and an average $\omega$ for the remaining species (model 3); and (e) average $\omega$ for parasites feeding on living tissues and blood (*Co. hominivorax*, *D. hominis*, and *G. morsitans*) and an average $\omega$ for all other species (model 4). The likelihood estimate for the neutral model was compared to the likelihoods of each model in test (model 1–model 4) using a $\chi^2$ statistics. The possible evolutionary processes involved in sequence evolution (neutral: $\omega = 1$, purifying selection $\omega < 1$ and positive selection $\omega > 1$) were evaluated for the significant comparisons ($\alpha = 0.05$).

Phylogenetic independent contrasts analysis: to evaluate the effects of autocorrelation in feeding habits due to phylogenetic relationships among the species under study, we performed a phylogenetic independent contrasts (PIC) analysis. For the PIC analysis, the average MNE for the four replicates per species and the phylogeny based on *Marinho et al. (2012)* were used as inputs in the ape package of the R environment (*Paradis, Claude & Strimmer, 2004*). Each species was assigned a number corresponding to its feeding preference (0: obligate parasite and 1: necro-saprophagous).

## RESULTS

Candidate gene selection: From the 21 possible candidates, eight genes were amplified and resulted in a clear and unique amplicon with high qPCR efficiency: *Cyp6g2, foraging, Glutamate dehydrogenase, Jonah 65 aiv, Malvolio, PGRP-SC2, RPS6-p70-protein kinase,* and *smooth* (see Tables S1–S3 and supplemental text for details).

Gene expression analyses: we used the comparative *Ct* method with efficiency correction (Mean Normalized Expression or MNE; (*Simon, 2003*) to compare mRNA levels among Calliphoridae species with different feeding habits in four biological replicates. PCR efficiencies were obtained from the standard curve of each primer, ranging from 91 to 104% (Table S2).

To identify genes potentially correlated with distinct feeding habits or with species diversification patterns, subsequent statistical inferences were conducted using a two-way ANOVA. Statistical inferences and mRNA levels values are shown in Tables S5–S7.

Possible candidate genes could be classified into three groups according to the divergence in gene expression patterns among the different species, by sex and developmental stage: (i) genes with conserved mRNA levels among all species; (ii) genes with different mRNA levels in at least one comparison, and (iii) genes with different mRNA levels in which the strongest factor was feeding preference.

The comparisons within larvae demonstrated low variation in mRNA levels in all genes, except *Mvl*. For them, no statistically significant differences were observed among species that might be correlated with feeding habit ($P > 0.01$; Fig. S1). This finding indicates a high degree of conservation in mRNA levels of these genes in the larval stage. For the adult stage, no delimited and clear pattern was observed, with two genes (*Cyp6g2* and *Gdh*) showing no statistically significant differences in comparisons between sexes and among species. The remaining genes showed significant differences in at least one comparison, but without any particular correlation with sex or species.

mRNA levels of *Pgrp-SC2* and *for* varied within species only in adult females ($P = 0.0003$ and 0.002; respectively) and were conserved in males. A similar result was observed for *S6k*, but mRNA levels were conserved in adult males instead ($P = 0.01$). The difference observed in *Jon65aiv* in females was significantly correlated with feeding habit ($P = 0.009$) and in adult males there was a difference among species ($P = 0.006$). The gene *sm* was differentially expressed among the different species in both sexes (adult males $P = 0.001$ and females $P = 0.01$).

Linear correlation analyses were then used to test if evolution in gene expression levels of the candidate genes were correlated with overall genetic distances among species

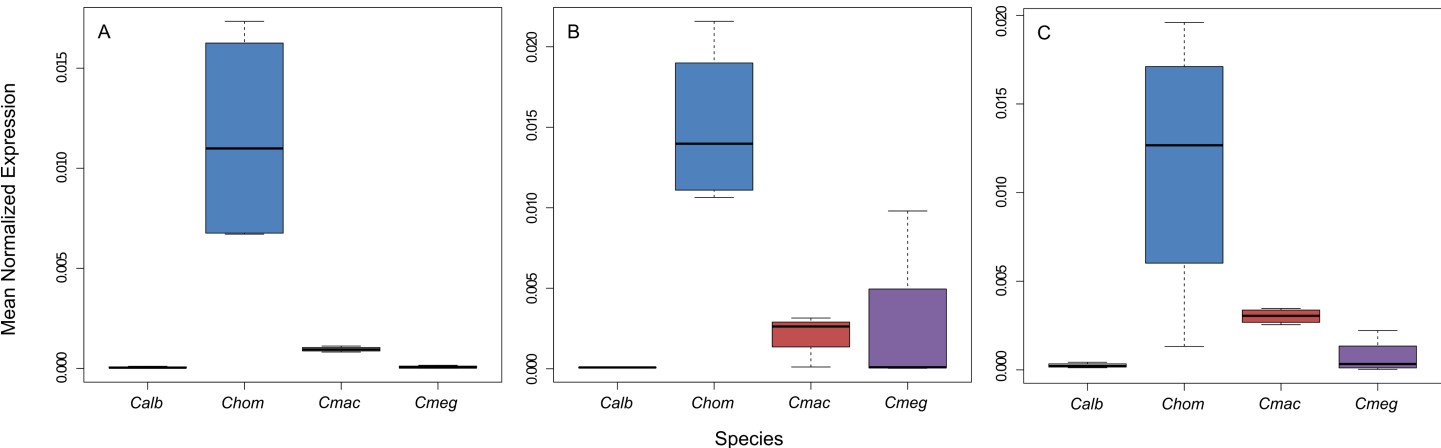

**Figure 1** **mRNA levels of *Malvolio* in larvae (A), adult females (B), and adult males (C).** The three boxplots show that the obligate parasite, *Co. hominivorax*, has a specific pattern of expression while necro-saprophagous species have a similar pattern among them. The highest expression difference observed was between *Ch. albiceps* and *Co. hominivorax* larvae. This gene was approximately 210 times more expressed in the obligate parasite. The smallest difference was among necro-saprophagous species. *Ch. megacephala* had 1.2 times higher expression than *Co. macellaria*. Calb: *Ch. albiceps*; Chom: *Co. hominivorax*; Cmac: *Co. macellaria*; Cmeg: *Ch. megacephala.* .

(neutral sequence evolution represented the null hypothesis). Pair-wise nucleotide sequence distances were calculated and correlated with the pair-wise gene expression divergence. None of the genes showed a significant correlation after the application of multiple testing corrections.

Interestingly, among all our candidate genes, only *Malvolio* showed a significant difference in mRNA levels between feeding habits in larvae ($P = 0.00002$) and both adult males and females ($P = 0.0007$ and $P = 0.00003$, respectively). The *Mvl* gene is expressed at higher levels in species with parasitic habit, while the necro-saprophagous species share a common pattern and level of gene expression, revealing a habit-specific expression pattern (Fig. 1).

These analyses, however, do not account for the evolutionary dependence among the species. Hence, we used PIC analysis to include the phylogeny of the chosen species and correlate feeding habit with mRNA levels. To this end, we corroborated our previous result that *Mvl* is somehow related to feeding habit (larvae: $r^2 = 0.9929$, $P = 0.01$). Surprisingly, *Gdh* had a similar significant result ($r^2 = 1$, $P < 0.001$ for all comparisons), indicating that it might be related to a feeding habit trait but is not specific to parasitism as shown previously.

Evolution of gene expression and coding sequence: Expression divergence (all genes) and coding sequence data (five of the candidate genes - see Methods section for details) were used to infer trees. Evolution of each gene was considered to be independent and tested against a standard neutral model using the phylogeny proposed by *Marinho et al. (2012)*.

Dendrograms (trees) constructed with mRNA level differences for all developmental stages and sexes showed a complex pattern among species. Dendrograms were built for each sex and developmental stage separately. A clear pattern relating gene expression of
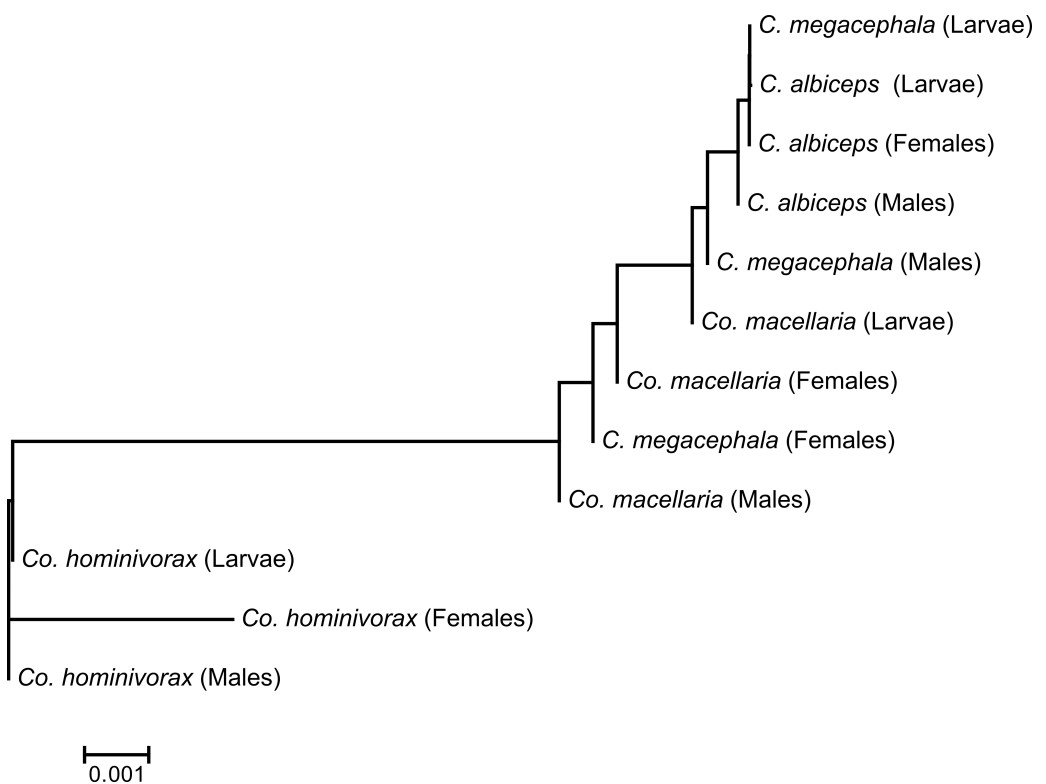

**Figure 2** **Dendrogram of *Mvl* expression using qPCR data.** The diagram shows the separation of *Co. hominivorax* from the necro-saprophagous species. This finding corroborates the idea that divergence in expression of *Mvl* is associate with feeding behavior. The scale bar represents the branch length estimated by expression divergence.

the *Mvl* gene with feeding habits was observed, with obligate parasite species separated from the necro-saprophagous species (Fig. 2). Indeed, feeding behavior was the major determinant of mRNA levels divergence, supporting previous statistical analyses.

For five of the candidate genes, we also inferred phylogenetic trees using coding sequence data. In all cases, neither the maximum likelihood (ML) nor the Bayesian inference (BI) methods showed deviations from the expectations under a neutral evolution model (i.e., grouping congeneric species together). However, results of the Tajima's Relative Rates Test (Table S8) showed a significant deviation from neutrality for at least two genes, *Mvl* (in both comparisons using different outgroups) and *for* (only when *Ch. megacephala* was specified as the outgroup taxon).

Further, we generated a phylogenetic tree adding six more Diptera species (parasites and non-parasites). All obligate parasites except *D. hominis* have a longer branch (the branch length represent the amount of genetic change) when compared to non-parasitic species (Fig. 3). The longer branch belongs to *G. morsitans*, the only species with a 69-bp indel in its sequence. Another notable observation is the clustering of *D. hominis* with *Cochliomyia* species.

To expand our knowledge of the evolutionary processes involved in *Mvl* coding sequence, we estimated the rate ratio of synonymous to non-synonymous substitutions per site using

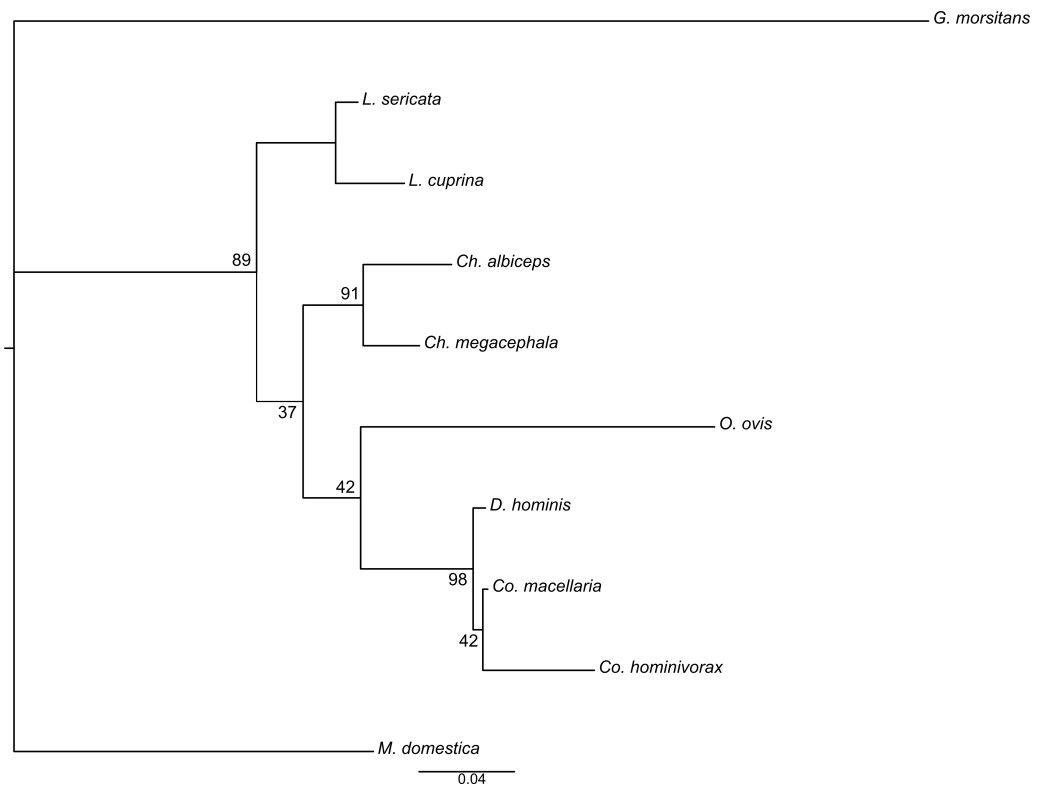

**Figure 3 Dendrogram of the *Mvl* gene including a broader range of species.** Species include obligate parasites *Co. hominivorax*, *D. hominis*, *L. cuprina*, *G. morsitans*, and *O. ovis*, and non-parasites *Ch. albiceps*, *Ch. megacephala*, *Co. macellaria*, *L. sericata*, and *M. domestica*. The majority of parasitic blowflies have longer branches compared to non-parasite species. The scale bar represents 0.04 substitutions per site.

the sequence and phylogeny of ten Diptera species (Table S9). First, we estimated the average $\omega$ for the whole tree and compared its likelihood with the tree generated from expectations under a neutral model (fixed $\omega = 1$). The result was statistically significant, allowing us to reject the null hypothesis in favor of a model in which this gene is evolving under a strong purifying selection ($\omega = 0.027$, $P < 000.1$). To test for differences in the selective processes constraining the tree, we performed the same test considering specific branches (see Methods). We found an elevated $\omega$ rate ratio in the *Co. hominivorax* branch compared to all other species (0.111 and 0.023, respectively), indicating relaxation of constraint at this branch.

Together, these results indicate that *Mvl* is associated with feeding preferences among Calliphoridae species, largely due to differences in mRNA levels and also with some influence of coding sequence variation. The molecular mechanisms of these sequence variations are probably located in specific sites (i.e., small segments of the coding sequence and/or regulatory region), thus accounting for the discordance between the coding sequence tree and gene expression data.

## DISCUSSION

A candidate gene approach was used to study the evolution of feeding habit in Calliphoridae. Messenger RNA levels of eight candidate genes were measured in larvae, adult males, and adult females of three necro-saprophagous species and one obligate parasite using quantitative PCR (qPCR).

A high degree of conservation in gene expression (transcript) levels was found in larvae of the four species. This suggests that regulatory variation in these genes is constrained by purifying selection and that these genes have an important function in development. The pervasive role of purifying selection has been described previously in genome-wide reports of various organisms, including mammals and flies (*Brawand et al., 2011*; *Nuzhdin et al., 2004*).

Some genes appear to be under different evolutionary selective pressures during different developmental stages. One of the most promising candidate genes, *for*, encodes a cGMP-dependent protein kinase (PKG), which participates in the modulation of the division of labor in social insect species and foraging behavior in *Drosophila melanogaster* (*Ben-Shahar, 2005*; *Fitzpatrick & Sokolowski, 2004*). This gene was differentially expressed among species in adult females, but neither in males nor larvae. This difference between sexes and stages requires the presence of different regulatory modules and the action of different selection regimes in each module. There was a difference in gene expression among species in one sex but not in the other for *S6K*, *Jon65aiv,* and *Pgrp-sc2* as well. Different phenotypes mediated by the presence of different regulatory factors has already been observed in stickleback fish (*Chan et al., 2010*). Chan and colleagues documented that the deletion of an enhancer in the gene *Pitx1* causes a contrasting variation of the pelvic skeleton between stickleback populations by reducing transcript levels. The result of this mutation is the presence of a population with reduced pelvises; however, the expression of this gene in other tissues essential for organism viability (non-candidate housekeeping genes) is similar between populations. These findings suggest that different regulatory factors are involved in the regulation of specific and essential structures by *Pitx1* during the fish development (*Shapiro et al., 2004*).

To test the hypothesis that the evolution of expression of these seven genes (in each stage or sex) was predominantly neutral (i.e., that regulatory variation accumulates in the absence of selection), we tested measures of expression and sequence divergence. In a neutral scenario, divergences in expression should accumulate linearly with respect to time. Pair-wise nucleotide sequence distances between species were calculated using four genetic markers. No significant evidence of gene expression evolving under neutrality was found. Together, our observations suggest that candidate gene expression levels are not strictly evolving under a standard neutral model of evolution. These findings are consistent with several studies that have demonstrated that gene expression is, in large part, subject to selection (*Nuzhdin et al., 2004*; *Ometto et al., 2011*; *Rifkin, Kim & White, 2003*).

In this study, *Malvolio* was found to deviate from neutral expectations with respect to gene expression levels between habits in all comparisons. *Mvl* is involved in the gustatory signal transduction via metal-ion transport into neurons regulating the sensory

perception of sweet taste (*Ben-Shahar, Dudek & Robinson, 2004*; *Orgad et al., 1998*). In *D. melanogaster*, behavioral studies have revealed the association of this gene with food preference (*Orgad et al., 1998*). In contrast, in the honeybee *Apis mellifera*, *Mvl* can define specialized tasks in the hive. Forager bees have higher *Mvl* mRNA transcript levels and increased responsiveness to sugar, while nurse bees exhibit lower expression levels and decreased responsiveness to sugar (*Ben-Shahar, Dudek & Robinson, 2004*). Indeed, *Mvl* plays an important role in foraging and food preference. According to our expression data, the obligate parasite, *Co. hominivorax*, has higher *Mvl* mRNA transcript levels compared to the necro-saprophagous species. In addition, taking into consideration the phylogenetic dependence among species, this study demonstrates a remarkable correlation between feeding habit and *Mvl* expression level. Although no information is available for Calliphoridae species, the *MVL* protein may also participate in the transduction of the gustatory signal, and play a role in food assessment. Differences in the expression and/or sequence of this gene may, in turn, result in differences of preferences for specific substrates.

Analyses under a phylogenetic framework, constructing dendrograms based on expression level differences and phylogenies based on the nucleotide sequence of coding regions, provided additional evidence for a possible role of *Mvl* in feeding habit differences among species. In the dendrogram generated for this gene, almost all parasites had a longer branch than the necro-saprophagous species. Additionally, the parasite *D. hominis* clustered with the genus *Cochliomyia*. However, it was previously observed that the affinities of Oestridae species with the remaining Oestroidea lineages are more difficult to recover with confidence and the group is usually recovered with variable placements in the superfamily's phylogeny (*Marinho et al., 2012*).

The correlation of *Mvl* with feeding behavior may be attributable to differences in transcript levels between necro-saprophagous and obligate parasitic species, although there may be some influence by signatures of sites under selection or similar molecular evolutionary processes (as suggested by the Tajima's test statistics). We confirmed with $\omega$ analysis that *Mvl* is constrained by purifying selection, but it is important to note that this is just a part of *Mvl* coding sequence. We might find molecular evolutionary signatures in other portions of the coding region or even in regulatory regions of the gene as well. The *for* gene might also be under influence of these differences, but these are probably due to specific sites under selection in the coding region. No evident pattern or significant difference in expression level was observed for the *for* gene.

## CONCLUSION

In this study, we tested the hypothesis that variation in gene expression is directly involved in feeding behavior. To date, most previous studies in this subject have been based on phylogenetic information alone (*Stevens & Wall, 1997*; *Stevens, 2003*). The results of the present study represent the first and expanded effort to understand the evolution of parasitism in Calliphoridae combining functional and molecular evolutionary approaches. Our findings indicate that *Malvolio* may be involved in the evolutionary genetic regulation of Calliphoridae feeding behavior, and additional studies into the molecular mechanisms undergirding the adaptive evolution of this candidate gene are needed and underway.

## ACKNOWLEDGEMENTS

The authors thank Rosangela A. Rodrigues for technical assistance and Maria Salete Couto for maintaining fly stocks. In addition, the authors thank Professor Cláudio José Von Zubem, PhD (UNESP, Rio Claro campus) for providing *Ch. megacephala* stocks and Professor Ana Cláudia Lessinger, PhD (UFSCar, Sorocaba campus) for providing *Co. macellaria* stocks. The authors are grateful for statistical support provided by Bárbara Bitarello for PAML analyses. We thank Dr Erika Braga, Dr Rasa Bernotiene, and three anonymous referees for their insightful comments and suggestions that improved our manuscript.

### Funding

This work was supported by grants to TTT from Fundação de Amparo à Pesquisa do Estado de São Paulo (FAPESP, grants 2008/58106-0 and 2014/13933-8) and from the Brazilian National Council for Scientific and Technological Development (CNPq, 477335/2009-8), and by a grant to AMLAE from Fundação de Amparo à Pesquisa (FAPESP, grant 2009/51723-7). GAC and RDM were supported by a fellowship from FAPESP (2009/13463-3 and 2014/01600-4, and 2012/19987-7, respectively). MATM received a fellowship from CNPq and FAPESP (150441/2016-9 and 2012/23200-2, respectively). TTT received a fellowship from CNPq (307502/2011-2). The funders had no role in study design, data collection and analysis, decision to publish, or preparation of the manuscript.

### Grant Disclosures

The following grant information was disclosed by the authors:
Fundação de Amparo à Pesquisa do Estado de São Paulo: 2008/58106-0, 2014/13933-8.
Brazilian National Council for Scientific and Technological Development: 477335/2009-8.
FAPESP: 2009/51723-7, 2009/13463-3, 2014/01600-4, 2012/19987-7.
CNPq and FAPESP: 150441/2016-9, 2012/23200-2.
CNPq: 307502/2011-2.

### Competing Interests

The authors declare there are no competing interests.

### Author Contributions

- Gisele Antoniazzi Cardoso conceived and designed the experiments, performed the experiments, analyzed the data, wrote the paper, prepared figures and/or tables, reviewed drafts of the paper.
- Marco Antonio Tonus Marinho analyzed the data, wrote the paper, prepared figures and/or tables, reviewed drafts of the paper.
- Raquel Dietsche Monfardini performed the experiments, analyzed the data.
- Ana Maria Lima de Azeredo Espin contributed reagents/materials/analysis tools.
- Tatiana Teixeira Torres conceived and designed the experiments, analyzed the data, contributed reagents/materials/analysis tools, wrote the paper, prepared figures and/or tables, reviewed drafts of the paper.
### DNA Deposition

The following information was supplied regarding the deposition of DNA sequences:
Genebank: KU234347– KU234388.

### Data Availability

The raw data has been supplied as Supplemental File.

### Supplemental Information

Supplemental information for this article can be found online at http://dx.doi.org/10.7717/peerj.2598#supplemental-information.

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
