# Peer review of "Evolution of genes involved in feeding preference and metabolic processes in Calliphoridae (Diptera: Calyptratae)"

_PeerJ, doi:10.7717/peerj.2598_

## Round 0.1 · original submission · Major Revisions

Your manuscript has been assessed by four expert reviewers. Based on their reports, and my own assessment, I am pleased to inform you that it is potentially acceptable for publication, once you have carried out some essential revisions suggested by them. Please provide more details about feeding habit differences among the four species studied.

·

Basic reporting

This article seems to me interesting and well prepared. English is good and text is clearly to understand, figures are relevant (only text is too small in some of them). Introduction and background are well written. This investigation is a first step to study the evolution of feeding habits in some dipterous.

Experimental design

No Comments.

Validity of the findings

This investigation is a pilot study in combining functional and molecular evolutionary approaches in flies. This study was performed professionally.

Additional comments

I have found some abbreviations, which, I think, should be explained. For example L116 - cDNA, L248 Ct method.

L339 should be "for" (gene) instead of "For" (gene)

L244 "From the 21 possible candidates...." It would be interesting to have information about these 21 possible candidates. This information can be presented as supplement material.

Text is Fig.1 and Fig 3 is two small, it should be enlarged.

Reviewer 2 ·

Basic reporting

The manuscript highlights how candidate gene approaches can be useful in generating data supporting the genetic basis of behaviours, often a difficult task, and likely a plastic response. The article is well written, with few grammatical mistake. Some citations are formatted incorrectly throughout the manuscript (i.e. Z, 1965 should be Zumpt, 1965 etc). Data are available.

Experimental design

The experimental design was well thought out and comprehensive - with just a single but VERY important question that was unanswered: for the rearing of the insects (larvae) for RNA extraction, what was their timeline (when were they removed and extracted, under what conditions), and on what they were fed? Were all 4 species fed the same substrate? Based on the reference for the rearing conditions, the C. hominivorax are reared on a different substrate. What evidence supports that the substrate is not what is generating the response seen (see additional comments below)?

Validity of the findings

The data are robust and statistically sound. However, more discussion into some of the variability observed (with the other genes, for example, sometimes you saw a great deal of variation in expression, other times you did not). See comments above for the experimental design, if the flies are not reared on the same substrate (as larvae), then the results observed could simply be due to the different substrate? Did you reciprocally rear the other species on the same substrate as C. hominivorax? It is possible that these questions are easily answered, but without knowing the exact rearing conditions just prior RNA extraction, the conclusions may not be supported by the data (i.e. it might just be a substrate effect and not an actual functional response).

Additional comments

As I am familiar with these species, it does not bother me much, but the shorthand species names should be addressed using 2 letters for genera rather than one, since they differ: Co. macellaria and Ch. megacephala. Using C. for both indicates they are the same, when they are not.

Reviewer 3 ·

Basic reporting

Sufficient. See General Comments for the Author

Experimental design

Sufficient. See General Comments for the Author

Validity of the findings

Sufficient. See General Comments for the Author

Additional comments

This report contains some novel findings that could be better explained, as noted in the following specific comments.


Specific comments
Abstract
Rewrite after amending the text.

Introduction
Line 91. Authors F and YZ?

Line 97. Reference the specific 'candidate gene approach' used by the authors.

Lines 102-104. What precisely is being corroborated? Reference the relative gene 'expression levels' approach to be explained more fully in the Materials and methods.

Materials and methods
Lines 113-114. Give search parameters.

Lines 114-115. Give design constraints.

Lines 117-118. Reference the geographical origins of each colony.

Lines 118-120. State the likely gut contents of ?-instar larvae and adults.

Line 136. Name and reference the thermocycler(s) used.

Lines 150, 155. Justify use of 'expression levels'. What's the turnover and stability of the candidate mRNAs?

Lines 159-161. Explain these hypotheses when mRNA quantities do, or do not, represent expression levels.

Line 195. Give alignment parameters.

Results
Line 245. Define 'without ambiguities'.

Line 260. Which larval instar(s)? Many moulting larvae?

Lines 255-299. Present each set of results using a more precise definition of 'expression'.

Line 304. Are there alternative explanations for finding a significant deviation?

Lines 322-326. Should this be in the Discussion? Has causality been demonstrated, or should you write Mv1 is associated with [not influences] feeding preferences? What is meant by 'probably located in specific sites'?

Discussion
Amend taking into account your definition of 'expression levels' and those in similar research reports that have been published.

Conclusion
Lines 399-400. What is the suite of findings that would support the hypothesis, and what is the suite of findings that would reject it?

Figures 2 and 3. Show statistical support for each node.

Reviewer 4 ·

Basic reporting

No Comments.

Experimental design

Well written. No comments.

Validity of the findings

Four other obligate parasites blowflies can be studied for the Malvolio expression level.

Additional comments

This paper attempts to reveal genes involved in feeding habit preference among blowfly species. The authors study expression levels of eight candidate genes that might be involved in the behavior and find that the expression level of a gene, Malvolio, differs among species and suggest that this gene is related to feeding preference. The aim of this study is interesting to understand the genetic basis of speciation of feeding behavior. However, their findings offer no information on the function of Malvolio. They select only eight genes for their studies, but this selection is not fair. It is unfortunate that they could only amplified eight genes among 21 candidate genes. Genes coding for chemosensory proteins (receptor and binding proteins) are good candidates. Then a systematic screening targeting for more numbers of genes would be the best approach.

1. The authors should describe details about the feeding habit differences (necro-saprophagous vs. obligate parasite) among four species. In particular they should explain how the feeding habitat of Cochlipmyia homonivovax is different from others.

2. It is surprising that the expression level of the Malvolio gene is highly elevated in one blowfly species. But the dendrogram based on this exceptional one case is not informative. I strongly recommend the authors to test other four species of obligate parasites for the Molvolio expression level.

3. The title of this manuscript is misleading as it is not clear that genes studied in this paper are involved in the feeding preference.

4. Malvolio is an ion transporter. The authors may discuss why the transporter is involved in feeding preference.

---

## Round 0.2 · accepted · Accept

The authors have properly addressed all the comments raĆ­sed by the four expert reviewers.

·

Basic reporting

I think that authors corrected and improved the manuscript.

Experimental design

No comments.

Validity of the findings

This is one of the first studies investigating gene expression in dipterous insects.

Additional comments

No comments.

Reviewer 2 ·

Basic reporting

All comments were appropriately addressed.

Experimental design

All comments were appropriately addressed.

Validity of the findings

All comments were appropriately addressed.

Additional comments

All comments were appropriately addressed.

Reviewer 3 ·

Basic reporting

The authors have made all the suggested amendments or successfully rebutted them.

Experimental design

No further comments

Validity of the findings

No further comments

Additional comments

No additional comments

Reviewer 4 ·

Basic reporting

No Comments

Experimental design

No Comments

Validity of the findings

No Comments

Additional comments

The authors responded to all the comments raised by reviewers and properly revised the manuscript text.